# Functional Precision Oncology: The Next Frontier to Improve Glioblastoma Outcome?

**DOI:** 10.3390/ijms23158637

**Published:** 2022-08-03

**Authors:** Dena Panovska, Frederik De Smet

**Affiliations:** Department of Imaging and Pathology, Katholieke Universiteit Leuven, 3000 Leuven, Belgium; dena.panovska@kuleuven.be

**Keywords:** functional precision oncology, glioblastoma, drug sensitivity

## Abstract

Glioblastoma remains the most malignant and intrinsically resistant brain tumour in adults. Despite intensive research over the past few decades, through which numerous potentially druggable targets have been identified, virtually all clinical trials of the past 20 years have failed to improve the outcome for the vast majority of GBM patients. The observation that small subgroups of patients displayed a therapeutic response across several unsuccessful clinical trials suggests that the GBM patient population probably consists of multiple subgroups that probably all require a distinct therapeutic approach. Due to extensive inter- and intratumoral heterogeneity, assigning the right therapy to each patient remains a major challenge. Classically, bulk genetic profiling would be used to identify suitable therapies, although the success of this approach remains limited due to tumor heterogeneity and the absence of direct relationships between mutations and therapy responses in GBM. An attractive novel strategy aims at implementing methods for functional precision oncology, which refers to the evaluation of treatment efficacies and vulnerabilities of (ex vivo) living tumor cells in a highly personalized way. Such approaches are currently being implemented for other cancer types by providing rapid, translatable information to guide patient-tailored therapeutic selections. In this review, we discuss the current state of the art of transforming technologies, tools and challenges for functional precision oncology and how these could improve therapy selection for GBM patients.

## 1. Introduction

Targeted therapies hold the promise to eradicate cancer cells through the inhibition of specific oncogenic proteins [1]. The efficiency of this approach largely depends on the dependency of the cancer cells to the targeted pathway, meaning that the identification of eligible patients is crucial to achieve clinical benefits. Current clinical practice uses a variety of diagnostic approaches through which disease-specific biomarkers are identified to select the most appropriate patients. For instance, the identification of HER2 amplified or estrogen receptor-positive breast cancer anticipates favorable response to HER2-targeted therapy or hormone therapy [2], EGFR mutation in lung cancer predicts response to EGFR-targeted therapeutic compounds [3], while imatinib in Philadelphia chromosome-positive leukemia predicts a favorable outcome in that context [4]. The completion of multiple cancer genome projects and the installation of better, faster and cheaper methods for genomic interrogations over the past 15 years has led to a better understanding of the pathogenic mutations that are connected to various cancer types, and fueled the concept of precision oncology [5]. Indeed, precision oncology aims at identifying effective therapeutic approaches based on properties (biomarkers) that are specific to each patient’s tumor [6]. While the success stories highlighted above have now been around for more than a decade, the applicability of this one-on-one relationship between specific biomarkers and associated therapeutic responses has also faced many challenges and could only be exploited to a very limited extent across available cancer therapies. There are several reasons why the identification of a simple biomarker to predict therapy response is not trivial. For instance, inter- and intra-tumoral heterogeneity can greatly hinder the interpretation for a treating oncologist as multiple genotypes (with potentially divergent treatment sensitivities) can simultaneously populate the same tumor, thereby significantly affecting treatment efficacy. Moreover, the simultaneous presence of additional, potentially interfering genomic aberrations further complicates the interpretation of the relationship between a single biomarker and clinical outcome.

### 1.1. The Complexity of GBM

Glioblastoma (GBM) [7], still the most malignant primary brain cancer in adults [8], significantly suffers from the above described drawbacks. Already since 2005, the standard-of-care treatment of GBM includes a multidisciplinary approach combining surgery, ionizing radiation (RT) and chemotherapy. In spite of this aggressive approach, the median survival of GBM patients generally does not exceed 2 years [9]. This is caused by a combination of factors. (i) GBM is a highly infiltrative tumor, meaning that surgeons are commonly unable to resect the entire tumor, resulting in significant amounts of residual disease. In line with this, the extent of resection (EOR) has been identified as an important prognostic factor for GBM [10]. (ii) Targeting the residual tumor cells, primarily done by a combination of radiation therapy and temozolomide (TMZ), turns out to be extremely difficult: already in more than 50% of patients, progressive disease is radiologically observed even before finishing TMZ treatment (typically already within 3 months of therapy) [9]. This strongly suggests that large amounts of intrinsically unresponsive tumor cells were residing in the brain tissue even before starting therapy, which rapidly cause recurrence in GBM patients. Identifying more suitable and patient-tailored therapies that are accessible to the central nervous system (CNS) and are able to target a heterogeneous population of tumor cells therefore remains a major challenge in achieving clinical benefits.

To identify appropriate drug targets for GBM, large-scale sequencing programs were initiated to uncover disease causing genetic aberrations [11,12,13]. Over the past decade, several hundreds of GBM tumors have been sequenced within various consortia, uncovering complex and elaborate genetic alterations, including single nucleotide variants, focal or large chromosomal deletions and/or amplifications, and gene fusions [13]. For several of these genetic aberrations, drugs that target the affected cellular pathways have been developed, either in the context of GBM or other cancer types. Examples of such targets/pathways include receptor tyrosine kinases (e.g., EGFR, PDGFRA, VEGFR, MET) and various downstream intracellular signaling pathways (e.g., PI3K, AKT, mTOR, MEK/ERK) [14], hyperactive fusion proteins (e.g., TACC-FGFR and NTRK-fusions) [15], DNA repair (ATR/CHK1/CHK2, MDM2/4, PARP1, WEE) [16,17], and cell cycle regulation (CDK4/6) [11]. In spite of numerous clinical trials that were conducted to test the efficacy of these drugs against GBM, clinical results have been disappointing [8].

The failure of these trials could in part be explained by a lack of sufficiently precise selection procedures to enroll the appropriate patients that could actually benefit from the given therapy [18]. So far, such selection has primarily been based on bulk genetic analyses, where the presence of specific genetic aberrations was used as inclusion criteria for assigning appropriate therapy for each patient—an approach used for instance in the INSIGhT trial for GBM patients (NCT02977780). However, the complexity and interpatient heterogeneity of the genetic aberrations in GBM are so extensive that multiple interfering pathways are often simultaneously affected [11,12,13], making it largely unclear whether tumor cells of a particular patient would be responsive to a given therapy (even in the presence of the particular targetable mutation). On top of this, with the advent of single-cell sequencing methods, it turns out that the cellular composition of a GBM tumor is more complex than initially anticipated. Indeed, single-cell RNA sequencing (scRNAseq) studies showed that multiple of the TCGA-based tumor cell subtypes and a variety of stem cell-like states (i.e., neural progenitor-like, astrocyte-like, oligodendrocyte progenitor-like and mesenchymal-like) can be simultaneously present in a single GBM tumor [19,20,21] while containing multiple, often divergent genomic aberrations. Moreover, the various stem cell-like cellular states are plastic, meaning that they are interchangeable, a process that seems driven by stress factors caused by the environment of the cells. As such, stem cell-like cells are often more resistant to therapeutic perturbations. Therefore, instead of the initially anticipated subgrouping into 4 major subtypes [11,22], current insights suggest that GBM tumors harbor dozens of different tumor cell profiles, probably each requiring a specific therapeutic approach [7].

Finally, GBM tumor cells can also acquire de novo resistance upon therapy. Indeed, in an initially TMZ-responsive tumor cell population, resistance can easily be acquired by upregulating DNA-repairing enzymes such as MGMT or by inactivating the DNA mismatch repair (MMR) system, eventually leading to tumor recurrence [23]. At this point, a second surgical resection is often used as salvage therapy combined with other chemotherapeutic agents, such as lomustine/CCNU [24]. Additionally, in the recurrent setting, it would be highly beneficial to have better tools available to identify more suitable therapeutic options. All the above shows that identifying an appropriate therapy for each GBM patient, either in the newly diagnosed or recurrent setting, remains a daunting task. Being able to more precisely match particular therapies to the appropriate patients would not only significantly increase our ability to delay disease progression, it could also increase the success rates of clinical trials by more precisely identifying eligible patients.

### 1.2. Exceptional Responders across GBM Trials

In spite of the overall inability to treat GBM with durable clinical outcomes, clinical trials sometimes describe small groups of patients that did show a clinical response. For instance, in the multicentric INTELLANCE trial (NCT01800695), a small group of patients did experience clinical benefits from the treatment. Recurrent GBM patients that harboured an EGFR amplification were treated with a combination of TMZ and anti-EGFR monoclonal antibody coupled to a toxin (ABT-414; DEPATUX-M^®^) [25,26], These exceptional responders included one patient with a durable response beyond 40 months in addition to 4 and 9 patients with a reduction in tumor volume of 25–50% and 25%, respectively, out of a total of 60 patients that received this treatment [27]. The overall statistics of the trial were however insufficient to warrant approval by the regulatory agencies [27], but being able to identify those patients more carefully could have improved the outcome of the trial. Furthermore, molecular profiling of the patient samples was unable to identify an overall correlation between OS/PFS and EGFRvIII mutations, even though preclinical results from GBM cell lines and xenograft models showed high specificity and effectivity of the antibody–drug conjugate towards EGFRvIII and EGFR amplified tumor cells [28]. This is one of the many examples where a direct relationship between a genetic aberration and therapy response could not be confirmed in clinical practice, highlighting that more sophisticated assays may be required to achieve better therapy matching. In addition, the molecular analyses in this and most other trials remains largely confined to bulk analyses without taking tumor heterogeneity into account. 

### 1.3. Functional Diagnostics: Evolving from a Static to a Dynamic Interrogation of Cancer Cells’ Ability to Respond to Therapy

Major efforts are currently being put in matching specific (genetic) cancer features to drug responses [29]. However, in order to determine therapeutic efficacy across different patients and within a single tumor, as highlighted above, genetic information alone is often proven insufficient. Indeed, most studies only use baseline measurements in a ‘static’ setting (i.e., one snapshot prior to treatment), and intend to correlate the presence of specific genetic features to subsequent responsiveness to therapy. The simultaneous aberration of multiple cellular pathways, which can significantly interfere with each other, or for which multiple therapeutic options are sometimes available, make it difficult to predict the most suitable therapy. A *functional* interpretation [6] (e.g., what happens before and after cells are exposed to a certain therapy; what are the effects of the drug on the cellular state) on the other hand could provide *dynamic*, faster, more detailed and potentially longitudinal insights into the ability of cells to respond to therapy in a genotype agnostic way, although methods to do so remain difficult. In this light, approaches for assessing differential drug responses are gaining traction by which live tumor cells are ex vivo exposed to various therapeutic insults, while a chosen cellular response is carefully monitored—an approach coined *functional diagnostics, functional oncology or functional precision medicine* [6]. When monitoring the right features, such approach does not necessarily require a complete biological understanding, while still providing medically relevant insights (e.g., do cells respond to therapy or not, rather than why do they respond or not), allowing faster translation to a clinical setting. 

Functional diagnostics is, however, not a novel approach. Such strategy has been widely applied in other biomedical domains, such as infectious diseases where antibiogram screens are used to select the most appropriate antibiotic in a patient-tailored way. Still, translating such functional diagnostic assays to a cancer setting requires further amelioration and validation in order to become medically applicable. In this regard, we endorse functional diagnostic insights as a complementary component to conventional genetic, imaging (i.e., MRI, CT scans) and baseline pathological (tissue) analyses. Indeed, coming to a proper patient-tailored interpretation will require that the different levels of information (imaging, pathology, genetic and functional assays) are integrated into an overarching framework to steer clinical decision making. 

Overall, the goal of functional testing is to bring forward personalized medicine to patients diagnosed with complex disease entities, where treatment options are rather limited. In other words, functional tests ought to facilitate the matching of each patient to the most beneficial treatment. This being said, ex vivo drug exposure of freshly isolated tumor biopsies can directly inform on cell death, alterations in signaling networks, cellular phenotype and morphology or even tumor cell–tumor microenvironment (TME) crosstalk and adverse events in normal tissue. Certainly, the type of functional readout informing on tumor and non-malignant cellular vulnerabilities would largely depend on the mechanism of action of the given treatment. Typically, investigators rely on commonly available, FDA-approved therapies or drugs in clinical trials where dose-escalating studies where safety and tolerability of the therapy of interest has been already assessed and approved.

In the particular case of GBM, the functional screening method should not only be able to map each tumor in great detail—given GBM’s high degree of intra-tumoral heterogeneity [30]—it should also be able to track molecular responses to drug treatments. Currently, every patient diagnosed with GBM is profiled using a uniformed diagnostic procedure, consisting of MRI scans and “static” measurements of pathological trademark alterations, such as chromosomal rearrangements, mutational patterns and MGMT promotor methylation (Figure 1). Functional testing gives the opportunity to directly evaluate therapy efficacy, either in dissociated GBM samples or tumor tissue slices [31]. In order to predict tumor cell behavior in such a complex and dynamic system as the GBM/brain, one must first familiarize with the baseline features (mutational status, transcripts, proteins/protein modifications, metabolites) and interactions between these components (gene–RNA, RNA–protein, protein–protein) across various cellular states [6,32]. Ex vivo drug sensitivity towards a panel of therapies can then be measured by monitoring the direction and strength of evolution of these interactive signaling events in all (non-) malignant cells. Miniaturizing the assay, for instance with chip technology would enable testing of multiple treatment conditions, while still providing sufficient multiparameter resolution on phenotypic and functional changes. These results could then be used by a medical board to integrate the functional finding (e.g., a ranked list of therapeutic options from most to least active in the tumor cells of the investigated patient) with baseline features and clinical parameters, such as tumor size, tumor location, extension of tumor resection and drug tolerability. Once all data is integrated, the most suitable drug/combination could be selected for each, individual patient. As anticipated, this procedure can be repeated once the GBM tumor recurs (Figure 1).

### 1.4. Tools and Methods for Functional Diagnostics

The development of functional diagnostic assays strongly depends on the availability of representative cancer models that maximally capture the genetic and phenotypic features of patients’ tumor. So far, in vitro cancer research has been relying on so-called conventional cancer cell lines [33], which, although easy to use and representative in broader disease terms, have important limitations including: (i) lack of predictive value with regard to activity in clinical trials; and (ii) display of major and irreversible alterations in biological properties, such as gains and losses of genetic information, alterations in growth and invasive properties and loss of biomarker expression compared to the original tumor [33]. The growing body of evidence of heterogeneity, along with technological advances and platforms for drug development, steered pre-clinical research towards models derived from diseased individuals, such as patient-derived cell lines (PDCLs), patient-derived organoids (PDOs) and patient-derived xenografts (PDX). For GBM, an armamentarium of such models has been developed, although the installation of optimal readouts to assess drug activity in either of them still remains challenging (Figure 2).

For the initial identification and evaluation of drug targets, compound design and efficacy testing, patient-derived cell cultures and organoids provide an excellent platform to preclinically explore and evaluate pharmacological responses across individual tumors. Given the fact that such models more faithfully recapitulate features of the tumor-of-origin, drug screening across cohorts of such models, offers the identification of therapeutic options that can be immediately linked to particular features present in the identified models [33]. As such, PDCLs/PDOs are compatible with large-scale pharmacogenomic platforms, such as Cancer Cell Line Encyclopedia [34], Genomics of Drug Sensitivity in Cancer [35], and Cancer Therapeutics Response Portal [36] and finally, the Connectivity Map [37], a project integrated in the CLUE platform (https://clue.io/about, accessed on 2 July 2022), comprising extensive and continuously expanding connectivity maps of protein, RNA expression and/or morphological changes in cancer cells, as a response to drug perturbation in addition to drug repurposing library of FDA-approved drugs, clinical trial drugs and pre-clinical compounds. All these large drug-screening resources are invaluable for exploring and understanding the mechanisms of various classes of compounds, drug repurposing and matching (combinations of) genomic mutations with functional responses over time [38]. However, reliable patient-derived in vitro models (PDCLs and organoids) may also take significant time to develop (from several weeks to months) and can typically only be generated from a subset of patients (for GBM, this ranges from 30–50%), thereby making them less suitable as generic tools to determine appropriate treatment regimens within acceptable time frames (from days to few weeks). Additionally, patient-derived in vitro models often lack the presence of an appropriate extracellular matrix (ECM) and (immune) tumor microenvironment (TME) which may also skew cellular behavior away from its original phenotype present in the patient [39]. Similarly, long-term culturing and expansion often leads to clonal selection and loss of heterogeneity [40], reducing their representative nature. PDX models emerged as patients’ avatars—in vivo systems that closely mimic primary tumor biology and features. In this manner, PDX models are not only a powerful tool for preclinical drug development and testing, but also proven beneficial in providing clinically relevant information upon PDX clinical trials [41] and co-clinical trials. In co-clinical trials, mouse PDX models are established from tumor samples of each clinical trial participant and serve as personalized models for drug testing, from which the most appropriate therapy can subsequently be applied to the patient/donor [42,43,44].

Drug screening in PDX clinical trials were executed for various cancer types and solid tumors [41]. With this concept. it was confirmed that PDX models have the ability to predict trial responses, by evaluating predictive response biomarkers, map resistance mechanisms [41] and guide treatment decision making [33,42,43,44]. Patient-derived xenograft models for GBM are generated by direct transplantation of dissociated patient tumor material or tumor pieces. While a tendency for CNV-loss in heterotopic models has been suggested, orthotopic PDX models typically retain a close resemblance to the primary tumor [45]. Interestingly, studies confirm that the tumor-of-origin resemblance is highly dependent on the region from which the biopsy has been harvested, meaning that two PDX models generated from distinct regions of a single tumor could generate PDX models with dissimilar tumor subpopulations [45]. XENOGBM is a study currently evaluating the molecular analogy between the primary tumors of GBM patients and their corresponding PDX models (NCT02904525). PDX platforms are more advantageous over in vitro cultures as they retain 3D structural organization, clinical features, such as tumor invasiveness, vascularization, pseudopalisading necrosis and therapy-induced tumor evolution, and molecular features of the primary tumor, for instance crucial biomarkers such as EGFR expression, which is regularly suppressed by culturing conditions [46,47]. Furthermore, orthotopic PDX models provide the in vivo CNS environment enclosed behind the blood–brain barrier (BBB) allowing the direct evaluation of the penetration capacity and metabolomics of pharmacologic agents. Although seemingly superior over other models, PDXs still have several disadvantages for precision medicine in GBM. These models are laborious, time consuming and expensive in comparison to cell lines and organoids. The tumor take rate has been shown to be quite variable, meaning that PDX models would not be generated for all patients, or the number of models would be too limited in order to evaluate sufficient numbers of drug or drug combinations. Furthermore, the time between tumor engraftment and therapy decision may be too long for GBM patients. Finally, the use of immunodeficient mice largely hinders the interrogation of the role of the immune system in treatment responsiveness and general tumor biology.

To circumvent these issues and in line with the rapidly advancing organoid technology, organoid cultures were successfully established from patient specimens or through pluripotent stem cell reprogramming. In either case, organoids represent self-organizing, 3D systems which highly resemble the tissue from which they were derived. PDOs conserve tumor heterogeneity and TME components, tissue architecture, molecular and functional features. PDOs can be efficiently expanded over time while conserving patient-specific genomic features and intra-tumor heterogeneity, which could be reliably correlated with functional responses to therapeutics [48]. Non-malignant cells that are retained after 2 weeks of culture in the GBM organoid models include macrophage/microglia, T-cells, stromal cells and oligodendrocytes [49]. These features make organoids remarkable platforms for high-throughput drug screening, treatment evaluation in personalized chemo- and immunotherapies [39] and prediction of patient outcome. As such, a growing body of evidence shows clear correlation between organoid in vitro responses to long-term clinical responses of individual patient donors. Currently, these evaluations have been mostly performed on patients diagnosed with gastrointestinal cancer [50,51,52], colorectal cancer [53,54,55,56,57,58], breast cancer [59,60], pancreatic cancer [57,59,61], ovarian cancer [62,63] and esophageal adenocarcinoma [64]. PDO conceptualization for personalized treatment is lagging for CNS tumors and needs further validation. PDOs were successfully propagated for 24 chordoma patients, enabling the evaluation of the response rate of PDL1-postive and -negative organoids to decreasing concentrations of nivolumab after 72 h [65]. Another study using PDOs for retinoblastoma (RB) tumors confirmed the therapeutic efficacy of a combination treatment of topotecan and melphalan against recurrent retinal tumors and subretinal seeds, which was in line with previous reports. Importantly, these RB PDOs contained tumor stroma consisting of glial cells, which have a tumor supportive role [66], again showing the potential of these models in precision medicine trials. GBM PDOs were successfully generated from primary patient tissue and allowed an in-depth characterization which confirmed the close resemblance to the patient material, not only on the phenotypic but also on the functional level [67]. However, GBM tumors are incredibly heterogeneous at the spatial level, so that PDOs derived from a single patient and three different tumor regions (infiltrating edge, necrotic core and bulk tumor region containing necrosis, gliosis and putative treatment response) generated organoids with functionally distinct features, implying a wide range of cellular diversity between the organoids [67]. How these differences influence therapy responsiveness is yet to be interrogated. Technical and methodological efforts are continuously being put into the improvements of GBM-PDO generation and maintenance. Thus, 4D-printed self-programmable cell-culture arrays were fabricated to alter and shape their 3D environment as a response to external stimuli, whereby the fourth dimension is time. As such, these platforms have been extensively used for characterization and high-fidelity drug screening purposes [68]. In addition, the time of GBM-PDO generation has been radically reduced: 1–2 weeks for 4D-printed models [68] and 2–4 weeks after surgery in a novel method, whereby micro-dissected tumor pieces are applied in an optimized, specifically formulated medium for GBM-PDOs propagation and placed on an orbital shaker (instead of Matrigel) [49]. This method avoids tumor dissociation and, in turn, enabled successful generation, biobanking, in-depth characterization and co-culturing with CAR T-cells, proving the specificity and capacity of CAR T-cells in targeting EGFRvIII mutant tumor cells [49]. To date, only one report noted the applicability of GBM-PDOs in guiding therapy regime, which was performed for a single patient. In this report, everolimus was selected as the most potent therapeutic drug among a panel of FDA-approved mTOR inhibitors and led to tumor regression in the patient diagnosed with a recurrent GBM [69]. As all previously described in vitro models, PDOs have some shortcomings, such as the lack of complete TME and vascular network of endothelial cells. Additionally, to maximally capture tumor heterogeneity, the patient material should be sampled from different tumor foci consisting of highly viable and metabolically active cells, avoiding necrotic and hemorrhagic areas. Obtaining such tissue from recurrent tumors can be challenging, because of the abundance of low-quality, cell-sparse and necrotic areas and lack of proliferating cells [49].

In general, tumors including GBM, release cells and cellular content into the bloodstream or cerebrospinal fluid (CSF). These biomarkers are shed from the tumor residing site in form of circulating tumor cells (CTCs), proteins, cell-free nucleic acids and extracellular vesicles (EVs), accordingly systemized as liquid biopsies [70]. As such, liquid biopsies set ground for a rapid, noninvasive way for cancer diagnosis and prognostic markers [70]. Currently, liquid biopsies have gained clinical application for metastatic breast cancer [71], small cell lung cancer [72], prostate [73] and colorectal cancer [74] in the context of tumor diagnosis and longitudinal monitoring of therapy responses in both primary and metastasized tumors. Specifically, it has been shown that CTC count in peripheral blood correlates to therapy response. Advanced molecular profiling of these cells shows a high degree of concordance between genomic and transcriptomic profiles with the tumor of origin, making CTCs an excellent tool that could support personalized medicine approaches [75]. CTC-derived cell lines for various cancer types enabled CTC characterization and in vitro drug treatments, which may inform on the treatment susceptibility of the primary tumor and identify ways to inhibit metastasis [76]. This has been further corroborated by short-term ex vivo propagation of small cell lung cancer (SCLC) CTCs in culture from 23 patients. The CTC-derived cultures were in vitro treated with cisplatin and etopside and the results were correlated with individual responses from the respective patients. The results of this investigation showed correlation between response profiles of ex vivo expanded CTCs and three patients. Furthermore, this study highlights the ability of in vitro treated CTCs to accurately inform on innate and acquired chemo-resistance, based on patients’ treatment history and clinical outcomes [72]. A similar report emphasized the predictive accuracy of in vitro-treated CTCs and two respective patients diagnosed with head and neck cancer and treated with cisplatin [77].

Current research is focused on refining methods for CTC isolation, ex vivo expansion and the establishment of CTC cell lines [78,79]. Additionally, two observational clinical trials, one in melanoma (EXPEVIVO-CTC; NCT03797053) and a second one in stage I-III lung cancer (CTMS 18-0056; NCT03655015), are anticipating the correlation of patients’ response to ex vivo expanded and treated CTCs. 

Owing to their location, brain tumors are challenging for surgical resection. Even when the tumor is accessible, the invasive surgery and biopsy collection present a risk of swelling and neurological damage. As patients receive an MRI scan within 12 weeks of treatment, contrast-enhancing lesions that are revealed on the images can indicate tumor progression, but might also be caused by post-radiotherapy edema, termed as pseudoprogression, which can spontaneously resolve [80,81]. At the moment, there are no methods that could reliably differentiate between glioma progression and pseudoprogression, or longitudinally monitor disease and treatment effects. The validation of biomarkers from liquid biopsies that could aid GBM prognosis is steadily progressing [81]. Liquid biopsies can be collected from cerebrospinal fluid (CSF), as it is in close contact with the CNS and accumulates tumor-specific markers, but CSF collection through lumbar punction is an invasive procedure [80,81]. In this light, the minimally invasive procedure to obtain liquid biopsies from GBM patients is through blood draw, but one must assume that the BBB is compromised at the tumor site. BBB disruption and permeability increases, as GBM tumors invade and progress into the surrounding tissue [81]. Therefore, CTC enumeration or EVs detection can potentially complement current strategies for more precise prediction of GBM progression. At present, methods for optimal CTC isolation and detection are advancing for GBM [70]. Unlike other epithelial-derived cancers where strong surface expression of EpCAM is detected (such as breast, prostate cancer, pancreatic, colorectal and hepatocellular), RNA sequencing of GBM-derived CTCs revealed Wnt-activated stemness and enrichment of mesenchymal features [81,82]. Alternative methods for CTCs detection in GBM include: GFAP labelling, telomerase-based assay, FISH detection of aneuploidy of chromosome 8, CTC-iChip microfluidic platform, recombinant VAR2CSA Malaria Protein and hTERT-specific oHSV1 expressing GFP [81]. All these studies point out the clinical utility of CTCs and liquid biopsies in real-time disease monitoring, prediction of progression and even functional measurements [83]. However, the number of CTCs is genuinely low (1–10 cells per 10 mL blood; 1 cell per 10^9^ blood cells); therefore, efficient CTC isolation which recapitulates intra-tumoral heterogeneity and enables functional assessment is still far beyond the reach of GBM patients [70,80,81].

Hence, the ideal model for rapid functional assessment of drug sensitivity in GBM would be a system which maximally preservers the native cellular integrity [40] and interaction of the tumor cells with the microenvironment [46]. This includes ex-vivo drug treatment of tumor slices [31] or cellular suspensions of freshly dissociated patients’ biopsies within hours post-surgery. Regarding GBM’s extensive heterogeneity and invasiveness, one must consider sampling from distinct tumor regions in order to gain an “as close as possible” perspective of the therapeutic vulnerabilities of the invading cells that are remaining after tumor debulking. In a recent proof-of-concept study of a single GBM patient, tumor material was harvested and analyzed with single-cell RNA sequencing and scATAC-seq. The leftover patient material was orthotopically transplanted into mice, which were then treated with standard-of-care therapy (irradiation and temozolomide). Subsequently, the patient tumor was harvested and analyzed at recurrence. This framework provided mechanistic genetic and epigenetic insights into therapy-driven evolution and identified potential druggable targets, therefore providing an approach for designing therapeutic regimens for GBM [84]. Yet another proof-of-principle study demonstrated the efficacy of drug screening human breast cancer cell lines through imaging mass cytometry, assessing more than 40 markers [85]. All these methodologies are facing technological challenges, which need to be improved, upscaled and validated in order to meet the needs of routine clinical practice.

An auspicious high-throughput drug screening methodology has emerged with microfluidic devices [86]. The chip technology closely mimics the extracellular environment, which in turn is capable of generating 3D structures of cells. Such a device was designed to recapitulate the complex vasculature of the BBB and track the transport of nanoparticles to GBM spheroids. Analogous permeability measurements were performed on orthotopic xenografts through intravital imaging, which matched the in vitro model [87]. Microfluidic devices are automated and multiplexed platforms where the controlled environment offers a way to monitor drug effects, such as cell viability, changes in cellular mass accumulation rate upon treatment and morphology [88] at multiple timepoints [86,89].

### 1.5. Clinical Trials Implementing Functional Diagnostic Assays

Currently, numerous clinical trials are testing the efficacy of functional diagnostic methods in the prediction of patients’ outcomes (Table 1). For instance, the EXALT-1 trial [90] showed that functional ex-vivo testing has the capacity to guide treatment and facilitate matching of patients with advanced hematological malignances to the right treatment. Strikingly, the progression-free survival in patients was prolonged 1.3-fold in comparison to the previously applied therapy. Briefly, patient material was obtained from biopsies, bone marrow aspirates or peripheral blood, dissociated (if necessary) and incubated with a drug library containing ~139 drugs at two different concentrations. After fixation, the cells were stained with antibodies against cancer cells and normal tissue, which allowed measuring the proportion of each population that remained alive following drug exposure. One of the greatest advantages of this approach was the short time between the testing and making treatment decisions [90]. A follow-up study, EXALT-2 is currently recruiting patients (NCT04470947) and in this three-arm study the treatment decision is going to be guided by genomic profiling, drug screening or the clinician’s choice. Whether this approach would favor direct, acute cytotoxic agents over slower-acting but potentially also very effective therapeutics options remains one of the outstanding questions.

For GBM, several clinical centers are engaging into functional measurements and ex vivo tumor profiling. At the Oslo University Hospital, GBM patients recurring or progressing after first-line treatment are recruited in the ISM-GBM study, in which individual cancer stem cells (CSCs) are first propagated in vitro as PDCL models, and subsequently subjected to high-throughput screening (HTS) towards FDA/EMA approved drugs (NCT05043701). An individualized drug combination would be prescribed to each patient, based on the outcomes from functional tests [91,92]. Similarly, the safety and efficacy of HTS in CSCs will be evaluated by the Swedish Medical Center (NCT02654964). While highly valuable as approach, the ability to propagate CSCs from GBM patients typically remains limited to ~30–50%, making this approach only applicable to a subset of patients [93].

A handful of preclinical studies have been initiated for GBM, where drug vulnerabilities of organoids or PDCLs are going to be measured in in vitro assays (NCT04868396, NCT04180046, PRISM—NCT03336931). In the case of NCT04180046, primary GBM cell lines are going to be established from patient samples, in order to pathologically characterize the presence of GBM-related hallmarks (IDH1, GFAP, P53, ATRX and Ki67) and measure dose-response effects of natural and synthetic drugs [94]. Similarly, the PRISM trial will perform detailed tumor molecular profiling of pediatric brain tumors on several levels (proteogenomic, transcriptomic, methylation analysis), which ultimately would enable treatment tailoring. In parallel, individual PDCLs and PDX models (“mouse avatars”) will be generated to facilitate the evaluation of the efficacy of the molecular-driven therapy within clinically acceptable timeframe [95]. The NCT04868396 study will, on the other hand, focus on the generation of organoid cultures initiated from tumor tissue collected by standard surgery. Here, the primary organoid library would be used to study mechanisms of aggressiveness and recurrence of GBM. 

3D-PREDICT (NCT03561207) is a multicenter prospective study which among other cancer types (ovarian cancer, advanced cancers) is also enrolling patients diagnosed with adult high-grade glioma (anaplastic astrocytoma and GBM). Here, extensive molecular profiling and direct ex vivo drug testing of patient tumor materials are carried out, with the ultimate goal being to make personalized medicine recommendations. Thus, the primary outcome measure of this trial is correlation between patient outcomes and functional results. Initially, four patients have been included in this study generating PDCLs and organoids for these patients. This enables the characterization and comparison of patients’ biopsies and individual tumor-derived models at genomic and transcriptomic level and further performing functional tests, such as clonogenic assays and 3D-PREDICT assays. The latter is practically a viability test of 3D spheroids treated with a mono-drug library at different concentrations for an adapted period of time, which extrapolates IC50 values and stratifies the response predictions as unresponsive, moderate and responsive. Based on these results, tumor spheroids derived from one of the four included patients (male, 24) diagnosed with GBM featuring ependymoma regions was found sensitive to JAK/STAT and mTOR inhibition. In this manner, clinicians opted to treat this patient with a combination therapy consisting of ruxolitinib and everolimus. After his seventh progression, the patient was classified as having stable disease for more than 4 months post-treatment [96]. A second study report from this trial enrolled 55 patients with newly diagnosed (ND) and recurrent high-grade glioma. In the case of ND patients, 71 patients were included at first. However, 13 patients had to be excluded because of model generation/assay failure. Then 15 patients were excluded because of premature enrollment in the study (<6 months); 9 patients decided not to take advantage of the trial and 1 patient progressed due to other events. As such, 33 patients were considered eligible for TMZ + RT treatment, where in-vitro TMZ response prediction was made 7 days post-surgery and subsequently compared to clinical OS after patients completed their treatment cycle. Of note, 20/33 patients had already progressed at the time this comparison was made. However, the median OS of assay responders was 11.6 months, as opposed to assay non-responders—5.9 months. Thus, 85% prediction accuracy was achieved. Interestingly, in the case of recurrent tumors, two remarkable observations were made. In some patients, PFS was exceeding the reported median PFS for carmustine and irinothecan and two patients were predicted as responders to BRAF inhibitors by the 3D Predict Glioma Assay, without harboring the targeted mutation, demonstrating the autonomy of in vitro/ex vivo tests in personalized medicine beyond NGS characterization. One of the patients diagnosed with GBM (IDH-WT), received combination of bevacizumab/dabrafenib and progressed after 4 months. The other patient had anaplastic astrocytoma and was prescribed dabrafenib for 12 months after which radiographical progression was noted. In both cases, at recurrence onset, patient tissue was collected by re-resection and screened in 3D Predict assay, recording a decrease in sensitivity towards BRAF inhibitors, which again was in line with the anticipated clinical outcomes [97].

An Ex Vivo DEtermiNed Cancer Therapy (EVIDENT) trial has been recently initiated (NCT05231655), which aims at determining the efficacy and feasibility of ex vivo screening in prediction of standard-of-care therapy outcome and novel therapy identification, including patients diagnosed with solid tumors (kidney, bladder, head and neck cancers, melanoma, sarcoma and GBM). This trial seems most prominent because solid tumor biopsies will be directly screened and the response would be quantified and correlated to patient clinical outcome.

**Table 1 ijms-23-08637-t001:** Clinical trials in GBM using functional diagnostic evaluation.

Identifier	Name	Title	Status	Models	Study Type	Purpose	Readout	Diagnosis (n = Number of Recruited Patients)	Ref.
**NCT05043701**	ISM-GBM	Individualized Systems Medicine Functional Profiling for Recurrent Glioblastoma (ISM-GBM)	Recruiting	PDCLs (CSCs from rGBM) *	Interventional	A personalized drug combination will be prescribed to each patient based on the functional drug screen	HTS FDA/EMA approved drugs; cell viability	rGBM (n = 15)	[91,92]
**NCT02654964**	/	Cancer Stem Cell High-Throughput Drug Screening Study	Unknown	PDCLs (CSCs from rGBM) *	Interventional	A personalized drug combination will be prescribed to each patient based on the functional drug screen	CSC/HTS viability assay of drugs/combinations	rGBM (n = 10)	/
**NCT04868396**	/	Patient-derived Glioma Stem Cell Organoids	Active, not recruiting	PDO	Observational	Baseline characterization	Mechanisms that contribute to aggressive tumor growth and treatment resistance in primary and recurrent GBM	ND-GBM & rGBM (n = 60)	/
**NCT04180046**	/	Utility of Primary Glioblastoma Cell Lines	Recruiting	PDCLs	Observational	Baseline characterization	Phenotypic, genetic (IDH-, MGMT- status) and IHC characterization	ND-GBM (n = 10)	[94]
**NCT03336931**	PRISM	PRecISion Medicine for Children With Cancer	Recruiting	PDCLs and PDX	Observational	Molecular profiling, drug testing, recommendation of personalized therapy	In vitro HTS testing; In vivo drug testing using PDX models; Liquid biopsies	Childhood solid tumors (n = 550)	[95]
**NCT03561207**	3D-PREDICT	3D Prediction of Patient-Specific Response	Recruiting	PDCLs and PDOs	Observational	Compare Assay results to reported patient outcomes	Cell viability	GBM, anaplastic astrocytoma/solid tumors (n = 570)	[96,97]
**NCT05231655**	EVIDENT	Ex VIvo DEtermiNed Cancer Therapy	Recruiting	Ex-vivo biopsies	Observational	High-throughput ex-vivo drug screen of cells processed directly from solid tumors to determine sensitivity/resistance profiles	Ex-vivo HTS of cells processed directly from solid tumors to determine sensitivity/resistance profiles	GBM, Solid tumors (n = 600)	/

* ND-GBM—newly diagnosed GBM tumor; rGBM—recurrent GBM tumor; HTS—high-throughput screening; CSC—cancer stem cells.

## 2. Concluding Remarks

Conventional genetic-based matching of patients and treatments may be beneficial for only a fraction of cancers, where the oncogene driver mutation is uniform and maintained at a stable level among the cancer cell population. Such examples are seen in HER2-positive breast cancers [2], Philadelphia chromosome in chronic myeloid leukemia [4], BRAF mutations in melanoma [98] and few other cancer types. However, this approach underestimates complex circuits of non-genetic mechanisms that define the pathological behavior of tumors. Therefore, most of the large-scale clinical trials matching targetable genetic alterations to inhibitors resulted in unsatisfying survival rates, widely accepting the fact that “one-size-fits-all” therapy approach is not beneficial for complex and heterogeneous diseases. From here, it became obvious that the personalization of cancer medicine is the way to tackle this disease. Although personalized chemo-sensitivity assays are thriving in the academic and commercial enterprises, still there are several hurdles that need to be addressed. Fundamentally, functional diagnostic assays require the availability of adequate tissue material to enable an efficient yield of viable tumor cells. This means that the hospital where the surgery is performed should include a department/laboratory, ensuring rapid transfer and minimal tissue manipulation before the functional diagnostic test is performed under strictly controlled conditions. Additionally, surgeons and clinicians should clearly communicate requirements and conditions for optimal tissue harvest and handling. However, in many cases, good-quality tumor samples cannot always be obtained, especially from metastatic and recurrent solid tumors [99]. Most commonly, core needle biopsies, fine needle aspirates and circulating tumor cells are collected, which are not sufficient for high-throughput ex-vivo drug screening or model establishment [100]. Next to sufficient viable cell yield, a key requirement for proper interpretation of functional diagnostic assays are treatment conditions. The diverse mechanisms of action of targeted inhibitors implies that concentrations and treatment duration should be optimized for each drug. One strategy to solve this is to evaluate various concentration and time ranges of each drug in representative cohorts of 2D patient-derived models/organoids and optimally validate a predictive biomarker correlating to response or direct measurement of tumor cell viability and fitness. Again, results from functional diagnostic assays should be routinely obtained within a clinically relevant timeframe. Considering GBM’s nature, all of the before mentioned points should be well considered. Firstly, based on the tumor location, surgeons are not always able to provide sufficient material for all pathological (IHC, genomic) and functional evaluations, meaning that a functional model and platform might not be established for all patients. In spite of GBM’s aggressive nature, the timeframe between the functional diagnostic readout to treatment selection should be well accounted. Finally, because of the vast spatiotemporal heterogeneity, biopsy materials sampled from distinct tumor regions might give rise in slightly biologically distinct models. Unfortunately, no current method can precisely profile remaining cell populations after tumor debulking, which eventually will invade the surrounding tissue and cause tumor recurrence. Efficiently targeting these cells remains a dreadful challenge for all oncology specialists. In summary, solemn genomic assessments do not identify obvious druggable targets and therapies for advanced and heterogeneous cancers. In this regard, functional diagnostic tests may provide a platform for exploring cytotoxicity profiles of cancer cells derived by affected individuals towards drug-and-drug combinations.

## 3. Future Directions

A growing appreciation of biobanking, the generation of living biobanks of patient-derived models and ex-vivo treatments have the potential to enhance the development of rationally selected combined therapies and guide prospective clinical trials. As the number of clinical trials and assessments of functional diagnostic platforms increase, we anticipate the implementation of this strategy in standard clinical oncology practice. Importantly, the integration of molecular characterization data, functional profiles, clinical parameters and patient follow-up from a multitude of individuals into a single database might even enable informing clinical decisions for patients from whom sufficient tumor material may not be available. Finally, while multiple endeavors are ongoing to implement functional diagnostics to select appropriate therapeutic options for GBM, it still remains to be seen which approach will prove to be the most predictive and clinically relevant.

## Figures and Tables

**Figure 1 ijms-23-08637-f001:**
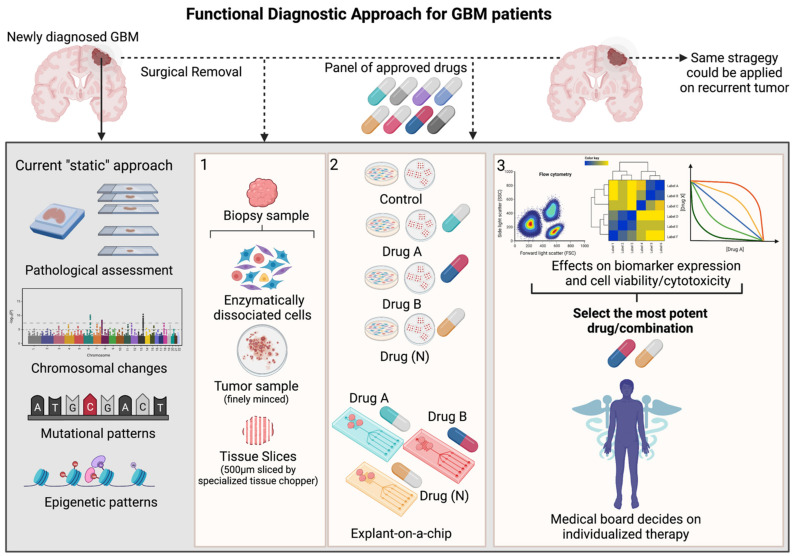
Schematic overview of functional diagnostic approach in GBM (Created with BioRender.com). During craniotomy, biopsy samples are routinely collected from newly diagnosed or recurrent GBM patients and pathologically assessed using standard clinical procedures, including immunohistochemical staining (IHC) of a handful of markers aiding histological grading, next generation sequencing and molecular analysis uncovering mutational patterns and epigenetic sequencing that measures the MGMT-promotor methylation status Although highly relevant, all these techniques offer only a single glance at the tumor’s baseline features (“static” measurement) and do not completely capture the intra-tumoral heterogeneity and therapeutic vulnerability of the patient’s tumor. To resolve this task, functional diagnostic is a personalized medicine strategy that makes use of live tumor samples derived from each individual patient. Panel 1: These biopsy samples can be enzymatically dissociated, minced or cut into fine tissue layers/slices. Panel 2: As such, these probes can be ex vivo treated with a panel of approved GBM-targeting therapies in cell culture flasks/plates or microfluidic chips. Panel 3: Various methods could be applied in order to optimally capture the effects of the given therapy on functional cellular features (cyto-toxic/-static events, various cellular states or cellular signaling pathways) relevant and corresponding to the given treatment. The output of these functional measurements would be a ranked list of most potent therapies, whereby a medical board could integrate this information together with histological, molecular measurements and clinical parameters. Finally, clinicians could decide on which therapy would be the most beneficial for each patient. This strategy could be applied on patients diagnosed with a recurrent tumor.

**Figure 2 ijms-23-08637-f002:**
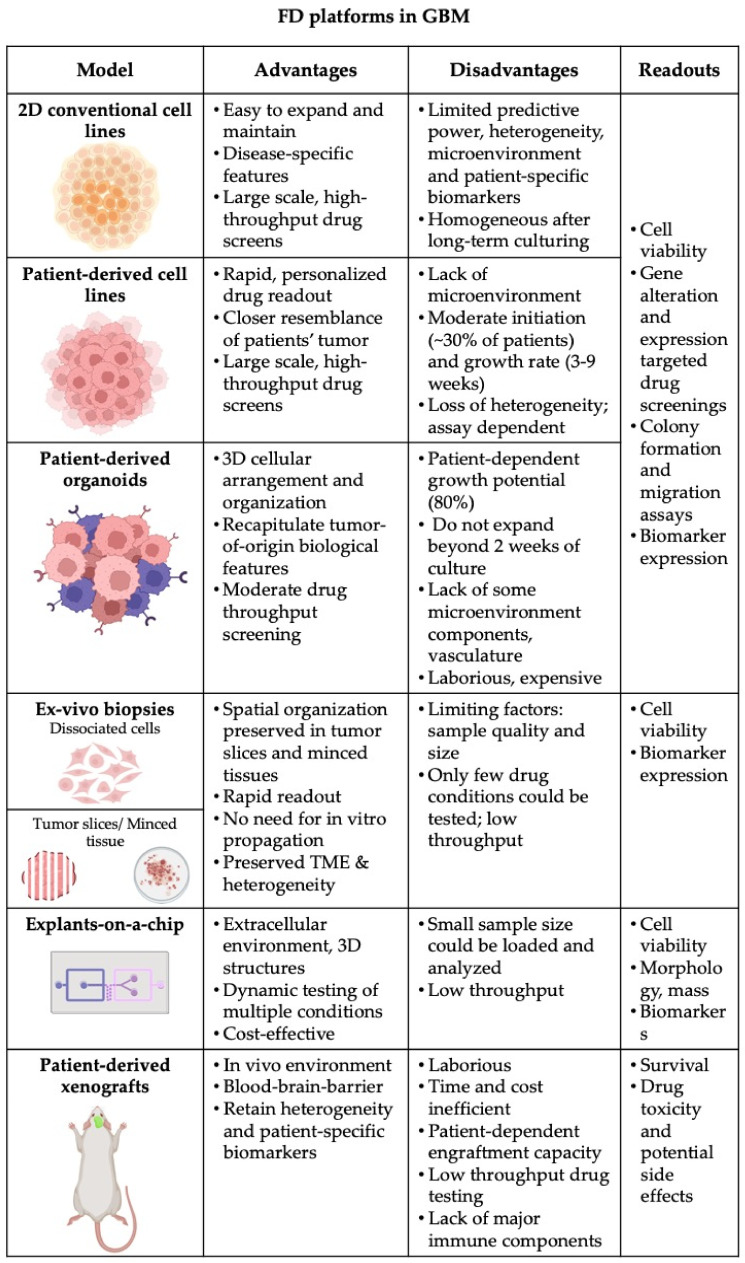
Summary of pre-clinical models and platforms, which could be used for functional testing in GBM. Advantages and disadvantages of each model together with potential assay readout are outlined (Created with Biorender.com).

## Data Availability

Not applicable.

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
