# Peer review of "Functional Precision Oncology: The Next Frontier to Improve Glioblastoma Outcome?"

_ijms, 2022, doi:10.3390/ijms23158637_

Round 1

Reviewer 1 Report

The well-written review article describes the methods for functional precision oncology and its relevance in glioblastoma multiforme (GBM). The authors raise attention to the importance of a legit subject and highlight the importance of considering the complexity of the tumor molecular profile for targeted therapy selection. Functional precision oncology uses a holistic approach to drug sensitivity predictions, regardless of knowledge on the molecular profile.

The introduction presents adequate literature review about precision oncology, the genomic complexity of the tumor type, methods of functional diagnostics with benefits and drawbacks, and catalogues clinical trials implementing functional precision oncology. In the discussion, the authors summarize potential use and obstacles of the approach in a logically structured format. In the future directions part, the authors mention the potential utility of database-associating clinical parameters and genomic results with functional diagnostic test results for future predictions in sample-deficit, which slightly contradicts to the need for a personalized attitude they highlight in the introduction. Apart from that, the review follows a coherent structure presents the topic from many aspects.

The article cites numerous related studies which are pertinent to the topic, and which support contextual understanding of the paper without excessive self-citations.

Major comments:

Drug testing on circulating tumor cells is not mentioned in Figure 2 and mentioned only in one sentence in the whole manuscript (lines 490-491), although it deserves further exploration (see for example PMID: 33777749). Please discuss this method in more detail.

Please address two additional issues in the manuscript:

1: It has been mentioned that functional diagnostics might prefer quickly acting over slower acting drugs. But also, can it also favor drugs with more adverse events (and more efficacy)?

2: How is it controlled during functional diagnostic experiments that the treatment does not affect normal cells?

The difficulty raised on the potential drug sensitivity disparity in different tumor regions (lines 320-324) is also applicable to drug testing on tumor slices, which the authors mention as an ideal example method of functional diagnostics. It would be important to mention this drawback in the paragraph around line 350.

Some minor comments:

It is not perfectly clear what is meant by the ’static’ and the ’dynamic’ differentiation. Line 150 „Indeed, most studies only use baseline measurements in a ‘static’ setting (i.e. one snapshot prior to treatment)…”. Taking a sample and doing functional diagnostics also represents a static snapshot before therapy. If dynamic is meant to reflect the potential to test several treatments in a row, then please discuss this a bit more. Please clarify the static and dynamic terms.

The sentence in lines 490-491 is arguable in general: “circulating tumor cells are collected, which are not sufficient for high-throughput ex-vivo drug screening or model establishment [85,86].” This might hold true for GBM, however many other publications prove the utility of drug screening using circulating tumor cells (CTCs) (e.g. PMID: 33207745, 33170321, 33932470 and also applied in commercially available services). Please, specify this sentence to GBM or write a bit longer about the method which uses CTCs.

Use of English is appropriate in general, however, please revise spelling of the manuscript. Some examples of typos: Line 46 „extend” extent, Line 86 „disappointung” disappointing, Line 162 “compete” complete, Line 445 „predication” prediction, Line 503 „for all patient” for all patients.

There is inconsistency in applying a space character before the citation reference (compare lines 372 and 473), please unify. Also delete extra spaces in lines lines 237, 267, 316, 385, and 419.

Please mind that the right border is missing from Figure 2 table.

Author Response

Comments from Reviewer 1:

  • Comment 1: Drug testing on circulating tumor cells is not mentioned in Figure 2 and mentioned only in one sentence in the whole manuscript (lines 490-491), although it deserves further exploration (see for example PMID: 33777749). Please discuss this method in more detail.

Response: We agree with this point. We have accordingly reviewed the suggested literature and discussed it in the manuscript.

  • Comment 2: It has been mentioned that functional diagnostics might prefer quickly acting over slower acting drugs. But also, can it also favor drugs with more adverse events (and more efficacy)?

Response: This is an important point, but we are not sure where in the text it is mentioned that faster acting drugs are preferred more than slower acting drugs. We only discuss that the time of readout would depend on the mechanism of action of the chosen drug. Some drugs would act faster than others and with functional diagnostics we would like to know which drugs are effectively targeting tumor cells.

This brings us to the question raised upon drugs with adverse effects. Normally, clinicians would use drugs that are available on the market, have been FDA-approved, or drugs that have promising results in clinical trials, where the safety and tolerability in human subjects have been assessed. In summary, a highly potent drug against tumor and normal cells, with more adverse effects, but acceptable safety/tolerability profile would be favored. Again, after a ranked list of most potent drugs is generated, it is up to the oncologist to decide whether to treat the patient with the suggested drugs generated by the functional platform or opt for treatment based on physicians’ choice.

Comment 3: How is it controlled during functional diagnostic experiments that the treatment does not affect normal cells?

Response: Thank you for this question. We discussed this aspect in the revised version of the manuscript.

  • Comment 4: The difficulty raised on the potential drug sensitivity disparity in different tumor regions (lines 320-324) is also applicable to drug testing on tumor slices, which the authors mention as an ideal example method of functional diagnostics. It would be important to mention this drawback in the paragraph around line 350.

Response: We agree with this comment and implemented it in the text.

  • Comment 5: It is not perfectly clear what is meant by the ’static’ and the ’dynamic’ differentiation. Line 150 „Indeed, most studies only use baseline measurements in a ‘static’ setting (i.e. one snapshot prior to treatment)…”. Taking a sample and doing functional diagnostics also represents a static snapshot before therapy. If dynamic is meant to reflect the potential to test several treatments in a row, then please discuss this a bit more. Please clarify the static and dynamic terms.

Response: Thank you for raising this matter. Indeed, a dynamic measurement refers to multiple time-course measurements or assessing sensitivity to a panel of drugs. We implemented this point in the text.

  • Comment 6: The sentence in lines 490-491 is arguable in general: “circulating tumor cells are collected, which are not sufficient for high-throughput ex-vivo drug screening or model establishment [85,86].” This might hold true for GBM, however many other publications prove the utility of drug screening using circulating tumor cells (CTCs) (e.g. PMID: 33207745, 33170321, 33932470 and also applied in commercially available services). Please, specify this sentence to GBM or write a bit longer about the method which uses CTCs.

Response: This is a very relevant topic in the field. We emphasized the potential of circulating tumor cells in the manuscript.

  • Comment 7: Use of English is appropriate in general, however, please revise spelling of the manuscript. Some examples of typos: Line 46 „extend” extent, Line 86 „disappointung” disappointing, Line 162 “compete” complete, Line 445 „predication” prediction, Line 503 „for all patient” for all patients.
  • Comment 8: There is inconsistency in applying a space character before the citation reference (compare lines 372 and 473), please unify. Also delete extra spaces in lines lines 237, 267, 316, 385, and 419.
  • Comment 9: Please mind that the right border is missing from Figure 2 table.

In addition to the above comments, all spelling and grammatical errors pointed out by the reviewers have been corrected.

Reviewer 2 Report

This review focuses on an attractive strategy based on functional precision oncology for glioblastoma patients. Albeit not new, this concept is gaining momentum as it is capable of providing important information that is not possible using static precision oncology approaches. In this light, the review is timely and relevant for a broader audience. The manuscript is very well structured, clearly written, and provides the reader with the necessary background as well as with overview of the state-of-the-art platforms for functional diagnostics. Additionally, updating the readers with clinical trails based on the functional diagnostics evaluation is another plus. 

Author Response

Thank you for giving me the opportunity to submit a revised draft of my manuscript titled Functional Precision Oncology to Advances in Molecular Genetics of Brain Tumors. We appreciate the time and effort that you have dedicated to providing your valuable feedback on my manuscript. We are grateful to the reviewers for their insightful comments on my paper.